# Stage II of Chronic Kidney Disease—A Tipping Point in Disease Progression?

**DOI:** 10.3390/biomedicines10071522

**Published:** 2022-06-27

**Authors:** Lovorka Grgurevic, Rudjer Novak, Grgur Salai, Stela Hrkac, Marko Mocibob, Ivana Kovacevic Vojtusek, Mario Laganovic

**Affiliations:** 1Department of Anatomy, School of Medicine, University of Zagreb, 10000 Zagreb, Croatia; 2Center for Translational and Clinical Research, Department of Proteomics, School of Medicine, University of Zagreb, 10000 Zagreb, Croatia; rudjer.novak@gmail.com (R.N.); salai.grgur@gmail.com (G.S.); stela.hrkac@gmail.com (S.H.); 3Department of Pulmonology, University Hospital Dubrava, 10000 Zagreb, Croatia; 4Department of Emergency Medicine, University Hospital Centre Zagreb, 10000 Zagreb, Croatia; 5Department of Chemistry, Faculty of Science, University of Zagreb, 10000 Zagreb, Croatia; mocibob@chem.pmf.hr; 6Department of Nephrology, Arterial Hypertension, Dialysis and Transplantation, University Hospital Center Zagreb, 10000 Zagreb, Croatia; ikovacevicvojtusek@gmail.com; 7Department of Nephrology, University Hospital Merkur, 10000 Zagreb, Croatia; mario.laganovic@gmail.com

**Keywords:** angiogenesis, chronic kidney disease, inflammation, proteomics, tissue regeneration

## Abstract

Chronic kidney disease (CKD) is the progressive loss of renal function. Although advances have been made in understanding the progression of CKD, key molecular events in complex pathophysiological mechanisms that mark each stage of renal failure remain largely unknown. Changes in plasma protein profiles in different disease stages are important for identification of early diagnostic markers and potential therapeutic targets. The goal of this study was to determine the molecular profile of each CKD stage (from 1 to 5), aiming to specifically point out markedly expressed or downregulated proteins. We performed a cross-sectional shotgun-proteomic study of pooled plasma across CKD stages and compared them to healthy controls. After sample pooling and heparin-column purification we analysed proteomes from healthy to CKD stage 1 through 5 participants’ plasma by liquid-chromatography/mass-spectrometry. We identified 453 proteins across all study groups. Our results indicate that key events, which may later affect the course of disease progression and the overall pathophysiological background, are most pronounced in CKD stage 2, with an emphasis on inflammation, lipoprotein metabolism, angiogenesis and tissue regeneration. We hypothesize that CKD stage 2 is the tipping point in disease progression and a suitable point in disease course for the development of therapeutic solutions.

## 1. Introduction

Chronic kidney disease (CKD) is the progressive loss of renal function that can develop rapidly over a period of months or advance more slowly over many years. It is recognized as a worldwide public health problem, with a global estimated prevalence of 13.4% and high associated healthcare costs [1,2]. The disease is caused by a diverse range of conditions (immunological, toxic, metabolic, etc.) that affect the functional kidney compartments, including vasculature, glomeruli and tubulointerstitium [3]. CKD is divided into five stages that are characterized by progressive deterioration of glomerular filtration rate. In the early stages, the symptoms are not specific, which can prevent correct diagnosis, and the disease is therefore often confirmed only in its more advanced stages, usually based on the glomerular filtration rate estimation (eGFR), determination of albuminuria and, at times, kidney biopsy [4]. The observed loss of function is progressive and irreversible due to renal circulatory impairment, which is common in patients with underlying diabetes, obesity, atherosclerosis or hypertension [5]. According to epidemiological and clinical data, as the kidneys deteriorate over time, the mutual endpoint is tissue fibrosis and dysfunction. The existing treatment options are non-specific and are aimed at improving the underlying condition. For patients in end-stage renal disease (ESRD), this includes renal replacement therapies, dialysis and kidney transplants, all procedures that require frequent follow-ups and are prone to complications [6]. In a study by Glorieux et al., the plasma proteome of patients with stage 2–3 and stage 5 CKD with haemodialysis confirmed changes in molecular pathways related to decreased haemostasis and increased inflammation, complement activation and vascular damage. An increase in plasma levels of two proteins was identified during CKD progression: lysozyme C and leucine-rich alpha-2 glycoprotein [7]. Romanova et al. proposed that serum alpha-1-antitrypsin and HSP90B2 are associated with ESRD and might be potential disease biomarkers [8]. To add, some authors focused their research on advanced CKD kidney derived proteins and proposed several proteins to be used as early plasma biomarkers of CKD, while others aimed to detect pro angiogenic and inflammatory biomarkers [9,10,11]. However, these results require verification, since the protein expression changes dramatically over the course of the disease stages, and it could therefore be misleading to estimate early biomarkers according to proteins expressed in ESRD. Gansevoort and de Jong argue that determination of proteinuria in stage 1 and 2 of CKD might be a relevant indicator of future progression to ESRD. Furthermore, as they argue that albuminuria might be an early marker of renal impairment—patients with detected albuminuria without eGFR impairment might benefit from cardioprotective and renoprotective therapies [12]. Although advances have been made in understanding renal disease, it is still difficult to pinpoint key events that underlay the advancement of kidney failure and identify therapeutic targets that could slow it down. Previous research in this field does not provide a clear understanding of the pathophysiology of CKD stages and fails to point out potential “turning points” that could signal irreversible organ damage. Up to now, early diagnostic protein biomarkers of CKD have not been extensively studied.

We performed a cross-sectional shotgun-proteomic study in order to screen the plasma of advancing kidney disease and compare them to healthy control participants. Molecular profiles of identified proteins were created for each CKD stage (from 1 to 5), with the aim to specifically point out markedly upregulated or downregulated proteins. Our results implies that the aggravation towards renal failure does not follow a “linear mode” of disease progression. We propose evidence of a potential tipping-point occurring at stage 2 of CKD, which possibly makes it a crucial pathophysiological moment that may significantly affect disease progression. Several signalling pathways were activated, some of which had previously been connected to renal pathology. Additionally, new possible plasma biomarkers of early CKD development are proposed for future validation. The presented data interconnects a plethora of the published research on CKD proteomics and hypothesizes stage 2 as a crucial event in disease progression.

## 2. Materials and Methods

### 2.1. Study Participants and Study Outline

This cross-sectional observational study included N = 90 participants divided by declining eGFR into the following experimental groups (N = 15): (a) CKD-1; (b) CKD-2; (c) CKD-3; (d) CKD-4; (e) CKD-5; (f) healthy controls (Table 1, Figure 1). Exclusion study criteria were: persons under the age of 18, patients with proven malignant, systemic autoimmune, rheumatic or central nervous system diseases, persons with mental and behavioural disorders, and patients with acute cardiovascular or infectious disease. The majority of participants with CKD suffered from underlying primary glomerular disease objectified by biopsy, including patients with stage 1 of CKD. Detailed distribution of underlying kidney diseases for each CKD stage is presented in Table 1. CKD was classified into stages using Kidney Disease: Improving Global Outcomes (KDIGO) 2012 nomenclature [4]. This study was approved by the Ethics Committee of the University Hospital Center Zagreb (EP-16/106-2, 24 June 2016). All participants signed an informed consent form. Blood samples from all participants were collected from 1 January 2017 to 31 December 2018 at the Department of Nephrology, Arterial Hypertension, Dialysis and Transplantation, University Hospital Center Zagreb.

### 2.2. Plasma Sample Collection and Heparin Column Chromatography

Blood samples were drawn into tubes containing 3.8% sodium citrate to form an anticoagulant-to blood ratio (*v*/*v*) 1:9. Plasma was obtained by centrifugation (15 min at 4 °C and 3000× *g*), and aliquots of each sample were stored at −80 °C until analysis. Samples were then pooled into six groups (CKD stages 1–5 and healthy controls). All pools were then applied twice through the 5 mL heparin sepharose column (HiTrap Heparin HP, Thermo Fisher Scientific, Waltham, MA, USA) and consecutively eluted with 5 mL of 0.5, 1 and 2 M NaCl solution. Eluates were then precipitated with acetone (previously cooled at −20 °C); (eluate:acetone ratio of 1:4) and were then left overnight to further precipitate at −80 °C. The following day, samples were centrifuged at 16,000× *g* for 10 min. Supernatant was disposed and formed pellets were resuspended with phosphate-buffered saline. Prepared pooled samples were then stored at −80 °C until further analysis. 

### 2.3. Liquid Chromatography-Mass Spectrometry (LC-MS) Analysis

Total protein concentration was determined using the RC DC Lowry protein assay (BioRad, Hercules, CA, USA) according to manufacturer’s instructions. Pooled protein samples (40 μg) were denatured (8 M urea), alkylated (55 mM iodoacetamide in 8 M urea) and digested with 0.8 μg of TPCK treated trypsin (Worthington Industries, Columbus, OH, SAD) in 10-kDa centrifugal filter units. Digested peptides were purified and concentrated using Stage Tips [13]. Tryptic peptides were then separated in a gradient of acetonitrile in formic acid on a PepMap C18 25-centimetre-long nano-column by high-performance liquid-chromatography on an Easy-nLC 1200 System (Thermo Fisher Scientific, Waltham, MA, USA). Mass spectrometry (MS) was performed on a T Q Exactive Plus instrument (Thermo Fisher Scientific, Waltham, MA, USA). Automated measurement cycles consisted of full MS scanning and MS/MS scanning of up to ten most intense ions. Full MS scans ranging from m/z 300 to 1800, were obtained in the Orbitrap analyser at a resolution of 70,000, with internal calibration of the instrument using the lock mass setting. 

### 2.4. Data Analysis

Raw data was processed using the Proteome Discoverer (Thermo Fisher Scientific, Waltham, MA, USA ) software version 2.4. Spectral data was searched using SequestHT and Mascot search engines in iterative manner against human proteome database retrieved from UniProt (Release 2020_05, Proteome ID: UP000005640). The search parameters were: full trypsin digest, missed cleavages max. 2, dynamic modifications: Met oxidation, Asn and Gln deamination, dynamic protein N-terminal modifications: Met-loss, acetylation, Met-loss and acetylation. Label-free quantification was performed to determine the relative protein abundances, which were calculated from MS1 peptide intensities and normalized between samples. Samples were analysed in technical duplicates, and proteins identified with at least one peptide were considered relevant for analysis. Obtained data were deposited at the ProteomeXchange Consortium via the Proteomics Identification Database (PRIDE) partner repository with the dataset identifier PXD033427. 

Gene enrichment analysis, including analysis of relevant biological pathways, was performed using Funrich 3.1.3. software which yielded several significant biological pathways [14]. Statistical significance among the biological pathways was determined designated as statistically significant for *p* < 0.05 by performing hypergeometric test, corrected by Bonferroni for multiple comparisons. A literature search for the statistically significant pathways was performed and we report the pathways we deem to be potentially the most biologically relevant by manual curation.

Individual identified proteins were initially analysed for outliers in a binary (yes/no) fashion for each group. Outliers (proteins found in a single experimental group) were then plotted as Venn diagram and for each outlier a literature search (described below) was performed.

Identified proteins were plotted in a heatmap according to their protein abundance per experimental group and analysed in a semiquantitative fashion. 

For each identified protein, an individual trend analysis and literature search was performed. Only heatmaps of proteins selected by manual curation are reported. Heatmaps were plotted using an open-source web tool—Morpheus [15]. Following initial protein selection, another round of search was performed in order to select the proteins we deemed to be potentially biologically relevant. 

A search of literature was conducted for each of the identified unique proteins in the CKD groups, where we also selected proteins with evidence pointing to their involvement in CKD by manual curation. Studies performed with human participants were deemed significant. Search of literature was performed using Medline, Web of Science and Google Scholar databases.

All data (not manipulated by manual curation) is reported in Appendix A or the PRIDE repository to ensure data transparency. Manual curation as a source of bias and semiquantitative analysis as a study limitation is further elaborated in the Section 4.

## 3. Results

### 3.1. Protein Identification

The study included plasma samples obtained from 90 participants (N = 15 per sample pool) with CKD stages 1-5 and healthy controls (Patient characteristics are summarized in Table 1 and Figure 1). Across all plasma samples purified on a heparin affinity column, a total of 453 proteins were identified by LC-MS and subsequent analysis by the Proteome Discoverer software (356 proteins in CKD-1; 353 proteins in CKD-2; 381 proteins in CKD-3; 375 proteins in CKD-4; 390 proteins in CKD-5). Common laboratory contaminants and proteins with less reliable identifications, i.e., false discovery rate over 1%, were not included in further analysis. Identified proteins are deposited to the PRIDE databases, and a descriptive summary of the identified proteins across groups can be found in the Appendix A. Multiple proteins had observable changes in their protein abundance trends across different CKD stages, as compared to healthy controls. Evidently, stage 2 of CKD had a unique profile in that we saw the most dramatic changes in this disease stage. The main clusters of selected upregulated and downregulated proteins are presented as heatmaps in Figure 2.

### 3.2. Functional Enrichment Analysis

In order to gain better insight into the relevant biological pathways activated in the diseased kidney tissues, and the proteins released to patients’ circulation, we performed a functional enrichment analysis on the obtained data set. All statistically significant pathways in the enrichment analysis are presented in Appendix A. Pathways selected through the enrichment analysis and manual curation (as described previously), as the most relevant were: β3 integrin cell surface interactions, innate immune system, complement cascade and lipoprotein metabolism. Relevant pathways are shown in Figure 3, including the number of identified proteins expressed as percentage of proteins identified from the total number of proteins involved in these biological pathways. A general trend can be observed, in which most of the pathways show a peak in the number of proteins involved in the initial phases of CKD, namely CKD 1 and CKD2; after which a gradual decrease until CKD 5 is noticeable. 

### 3.3. Outlier Analysis

Across all disease stages we identified 298 common proteins, however some were unique to specific disease stages: 7 were found in the CKD 2 group, 5 in the control and CKD 5 group, 1 in the CKD1 and CKD 4 groups, whereas no unique proteins were found in the CKD 3 group (Figure 4). Through manual curation we selected those possibly relevant for renal pathophysiology: chromogranin-A, sex hormone-binding globulin and vascular cell adhesion protein 1 (Table 2). These proteins were linked to CKD progression, fibrosis stage and disease activity in human studies.

## 4. Discussion

Renal tissue can be damaged by a variety of noxious stimuli, such as long-term high blood pressure or elevated cholesterol, infections, medications and underlying conditions like diabetes or kidney stones, as well as by other immunologically-mediated mechanisms. If untreated, the injury can progress from a largely undiagnosed stage 1, to the more pronounced disease stages (Figure 5). The progressing pathology of renal disease and its turbulent pathophysiological changes are also visible on a systemic level, through the plasma (proteome) of CKD patients. Early disease stages that are characterized by a rather mild loss of function (reduction in GFR) are mostly associated to initial scarring of the glomeruli, however, we have observed substantial changes in molecular patterns that occur as the disease begins to develop. 

To summarize our findings, we organized them into several biological processes according to the proteins involved in renal pathophysiology, that modulate inflammation, angiogenesis, tissue repair and lipid metabolism. Stage 2 of CKD seems to be a “tipping point” in renal response to injury—there were clearly observable changes in the plasma abundance of several proteins previously associated to the failing kidney. An overview of potential target molecules identified by our study and the corresponding significant biological processes is shown in Figure 5. These new insights support and summarize the current knowledge, and focus the discourse towards, what seems to be, the critical point in the natural advancement of renal failure. 

The long-term smouldering inflammation in a CKD-ridden kidney supports the secretion of inflammatory mediators and progressive activation of other parts of the innate immune system. Complement proteins can trigger the cascade activation of innate mechanisms that damage renal tissue and lead to autoimmune-initiated glomerulopathies [23,24]. We present evidence of several complement (C) activation mechanisms that were clearly disrupted, mostly in stage 2 CKD patients. This can be observed in our analysis of biological pathways, as it shows a peak in proteins related to the complement cascade in CKD 1 and CKD 2. Furthermore, as the levels of complement inhibitors like C factor H decrease, the stage is set for activation of the alternative complement pathway, through C factor B. At the same time, the levels of complement activator properdin drop, which complicates kidney repair and inflammation resolution, as shown in renal disease C3 glomerulopathy and lupus nephritis [25,26,27,28]. The pronounced complement activation flares can thus consume the available properdin leading to a drop in its plasma levels [29]. The complement cascade can also be activated via the lectin pathway, mediated by pattern recognition molecules mannan-binding lectin (MBL) and MBL-associated serine protease-2 (MASP-2), which contribute to the pathogenesis of renal isograft ischemic reperfusion injury [30]. Our study shows that plasma abundance of MASP-2 drops sharply at CKD stage 2 and remains consistently lower than in healthy control samples. This favours the deposition of C3 on the tubular surface leading to renal injury, a phenomenon that is implied by our find of elevated plasma C3 levels in the later CKD stages. Activation of the lectin pathway can also be aggravated by ficolins, which we have also seen in stage 2 patients [31,32]. Adding “fuel to the fire” around stage 2 CKD, the secretion of acute phase plasma proteins like hemopexin, haptoglobin and serum amyloid A worsen the inflammation [33,34,35]. Another puzzle piece in stage 2 is the exclusive detection of vascular cell adhesion protein 1 (VCAM-1), a biomarker of kidney diseases. VCAM-1 enables the adhesion of leukocytes to vascular endothelium, aggravating atherosclerosis and further facilitating inflammation [17,18]. We also observed that proteins related to innate immunity pathways are raised above healthy control group levels in all CKD stages, with a clearly visible peak in CKD2 (Figure 3). In conclusion, the synchronous activation of multiple inflammatory pathways, clearly coincide in the early disease stages. Interestingly, the rapid build-up of these mediators does not suffice for neutrophil-induced myeloperoxidase (MPO) secretion. MPO is the neutrophils’ “weapon of choice” against pathogens that can also damage host tissues. We, as well as other researchers have shown, that its plasma levels are significantly lower in CKD patients, possibly due to the inhibitory effect of uremic toxins on this enzyme [36].

Imbalance of angiogenesis regulators is a known aggravating factor of CKD [31]. We have confirmed previous findings of excessive plasma angiogenin (ANG) in the early stages of CKD, however they drop and normalize in the later disease stages [37]. ANG contributes to blood vessel normalization, i.e., proper covering of pericytes over leaky vessels, promoting cellular adaptation to kidney injury [38,39]. Another role of ANG is to mitigate blood clotting by stimulating the activation of plasminogen [40]. Lack of ANG corresponds to lower plasminogen levels—a finding we have also observed in all CKD patients, which is in compliance with previous research. The consistently decreased fibrinogen levels contribute to blood coagulation abnormalities, which are common in CKD [41]. Another angiogenesis modulator we detected is neuropilin-1 (NRP-1), a co-receptor expressed in kidney pericytes that regulates the integrity of glomerular basement membrane, directly influencing glomerular filtration [42]. It was reported that overexpression of NRP-1 could damage kidneys in lupus nephritis [43,44]. Knowing that, the sharp increase of its protein abdundance in stage 2 of CKD that we have seen, might contribute to the disease pathology. Moreover, we found that integrin signalling, namely β3 integrin, which has been shown to play an important role in angiogenesis, peaks in early stages of CKD [45]. Finally, we have found that plasma levels of ceruloplasmin, an angiogenesis inducer, also peak at stage 2 of the disease. Taken together, it is evident that the angiogenesis pathway is a major disrupted network that should be further investigated in the early CKD stages [46,47]. 

In parallel with inflammation and angiogenesis, the stressed renal tissue likely triggers defence mechanisms and production of mediators related to tissue repair. As podocytes recover from injury, they undergo actin cytoskeletal remodelling, a process tightly regulated by cofilin-1, which was previously proposed as a kidney injury biomarker [48,49]. As CKD progresses, we have seen a steady raise not only in plasma cofilin-1, but also in its related mediator, vitamin D binding protein (DBP) [50,51]. In inflamed and injured tissues affected by CKD, DBP induces inflammation through cellular immunity and is known to be elevated in the serum of kidney transplant recipients with acute rejection We have observed an increase of DBP abundance in all CKD stages, peaking in stage 2 patients, which was previously found by Doorenbos et al. However, Kalousova et al. observed significantly decreased levels of DBP in CKD patients compared to healthy controls [51,52]. Another tissue repair modulator, elevated in CKD, is pigment epithelium derived factor (PEDF), a serin protease inhibitor with anti-inflammatory, anti-thrombogenic and vasculo-protective properties [53,54,55]. Our data indicated its elevated plasma abundance levels during disease progression, which could be interpreted by plasma soluble PEDF providing an extra supply for the dwindling tissue levels of the modulator from systemic circulation, possibly playing a protective role against organ damage. Kidney inflammation might also be internally ameliorated the through the control of inflammatory cytokines. Alfa-2-macroglobulin (A2M) is known to be elevated in patients with nephrotic syndrome—we have also found its excessive abundance in the plasma of CKD patients, again peaking at stage 2 [56,57,58]. A2M plays an active role in quenching the inflammatory reaction and fostering tissue repair, since it binds and neutralizes multiple cytokines. However, as inflammation in stage 2 is evident, besides tissue repair modulators, we have seen evidence of excessive matrix deposition, possibly leading to tissue fibrosis, which as it progresses, leads to clinical deterioration of CKD (Figure 5). Amongst dysregulated extracellular matrix proteins, lumican acts as a pro-inflammatory signal that peaks in stage 2 CKD and remains high in the following disease stages [59,60]. Elevated lumican levels may be related to the progression of renal fibrosis, which is in correlation with previous research [61]. To add, the protein abundance patterns of fibulin-1 and extracellular protein 1 from our data are not in line with previously published data: we consistently documented their decrease in CKD. However, previous studies were exploratory kidney disorder screens that did not stratify CKD patients according to stages; furthermore, the proteins in question lack specificity pertaining to kidney pathology [62].

Caught between pro-inflammatory signals, disrupted angiogenesis and tissue repair pathways, the kidney suffers from loss of function, which is apparent in lipid metabolism. CKD is a known risk factor for cardiovascular disease, since elevated plasma levels of apoA-I containing lipoproteins may lead to the accumulation of atherogenic particles [63,64]. In our study, multiple apolipoproteins were dysregulated with respect to control samples. apoA-I, apoA-II and apoA-IV peaked at stage 2 CKD: the most striking build-up was seen in apoA-IV, which was previously established as an independent predictor of CKD progression [65]. Therefore, the peak in proteins related to lipoprotein metabolism pathways which we observed in stages 1 and 2, might reflect this peak in plasma apolipoproteins. 

Our study has several limitations which must be taken into account while interpreting our results: (A) One of the main limitations of our pilot proteomics study is the relatively small sample pool we draw our conclusions from: the presented ideas should be more broadly validated. (B) This study is cross-sectional in design, a prospective longitudinal study in the future might offer further validation. (C) Additionally, the staging of CKD relies on an attempt at quantification of kidney function through eGFR and it is blind to the underlying cause of kidney failure. Stratification of patients according to primary kidney disease in future studies should distinguish whether these concepts are universal or disease-specific. (D) Another important limitation is sample pooling, which was a deliberate choice that enabled us to focus on the most prominent pathways disrupted in CKD, however it inherently diluted out individual patient specificities [66]. Therefore, only the most prominent regulators of CKD progression might have been identified, while weaker, but possibly important modulators might have been disregarded. Sample pooling might have also clouded the distinction among underlying kidney disorders, however, the action of sample pooling allowed for a clearer data analysis. It is also theoretically possible that specific overabundant proteome pattern of a single patient acted as a potential confounder that “outweighed” group from the “biological average”. (E) To add, the fact that we purified our samples using a heparin-enriched column “blinded” us to one part of plasma proteome, however multiple growth factors and cytokines that have a relatively high heparin binding affinity, were thus concentrated [67,68,69,70,71,72,73]. (F) Finally, it is important to note that these results are primarily interpreted by manual curation done according to literature search results. Therefore, data interpretation is prone to human error and publication bias. Furthermore, the comprehensive nature of mass spectrometry-based proteomics, uncovered abundant differences in plasma proteome profiles of CKD patients as compared to healthy people; therefore, for the sake of clarity and readability, we have omitted to comment on pathways clearly known to be disrupted in CKD, such as the insulin like growth factor binding proteins [74,75]. All raw data is, however, available at the ProteomeXchange Consortium via the PRIDE partner repository to achieve maximum transparency, as mentioned in the Section 2.4.

In spite of the mentioned limitations, our study boasts a few strengths: we have systematically analysed gross plasma proteome profile changes, as well as each identified protein individually. Our study group design enables tracking of molecular progression from healthy controls to ESRD, across all stages of CKD. Furthermore, it allows for an easier bench to bedside translation, as CKD stage is most commonly determined by eGFR in the clinical setting. Another major strength lies in the fact that of our data is in line with a plethora of studies done by other researchers in the field, which were often focused on a single protein, it thus serves as a form of “inherent quality control” of participant sampling and methodology. Therefore, our data is consistent with multiple studies focusing on kidney pathology, but also uncovers previously unknown CKD mediators and brings forth several novel findings.

The key message of our study is the observation of synchronous activation of cellular pathways related to inflammation and angiogenesis as well as tissue homeostasis and repair. In that regard, stage 2 seems to be a potential “tipping point” in CKD progression, a possible molecular milestone in natural disease progression, beyond which, hallmarks of disease, such as hemodynamic changes, proteinuria and inflammatory processes become increasingly evident both clinically and by proteomic analysis (Figure 5). Different biological processes known to be important for CKD progression are reflected in several proteins identified as potential target molecules (Figure 5). We thus hypothesize that stage 2 might be the potential “site” for biomarker discovery and for development of therapeutic strategies for halting CKD. However, further prospective studies with longitudinal design, including more participants and with adequate patient stratification are needed to confirm our hypothesis. These studies should, therefore, be the basis of further biomarker screening and future validation, which might ultimately have an impact on therapeutic strategies and potentially improve patient outcomes.

## Figures and Tables

**Figure 1 biomedicines-10-01522-f001:**
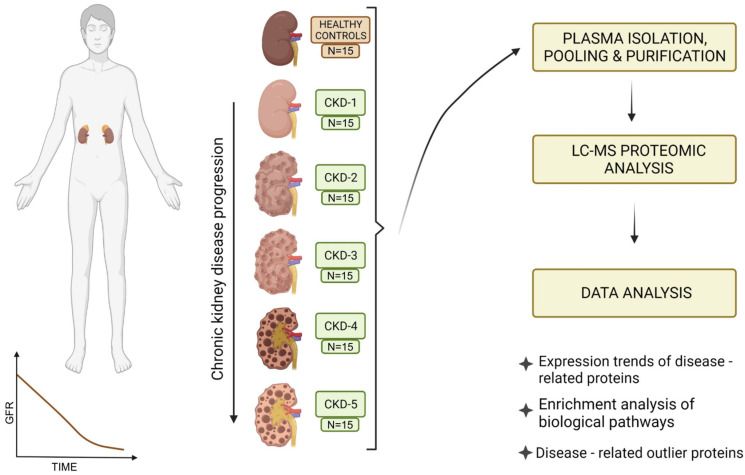
Study outline. Proteomic data was obtained from the pooled and purified plasma samples of patients with chronic kidney disease (CKD) including all stages from CKD 1 to CKD 5. The most relevant proteins related to kidney disease initiation and progressions were selected by manual curation, as well as kidney specific protein analysis was performed. Next, dynamic profiles of proteins identified in all samples, as well as outliers, were performed with an emphasis on the degree of renal failure at which the greatest changes were observed. Functional pathway analysis of proteins included in the ascending and descending protein abundance profiles was assessed.

**Figure 2 biomedicines-10-01522-f002:**
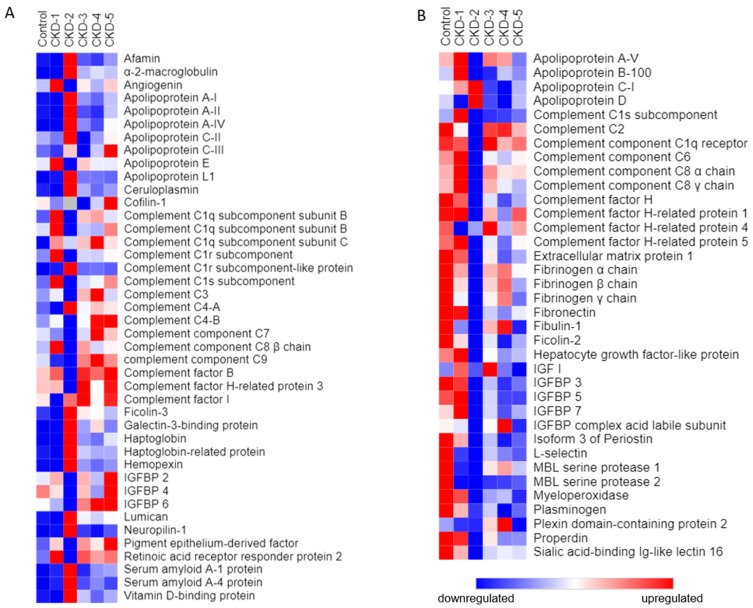
Heatmap showing selected proteins which were observed to generally have: (**A**) an increasing trend through CKD stages 1–5; (**B**) a decreasing trend through CKD stages 1–5.

**Figure 3 biomedicines-10-01522-f003:**
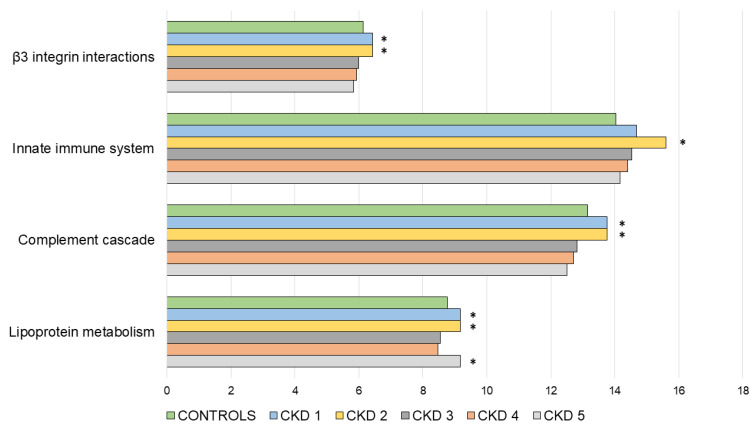
Bar charts showing relevant biological pathways (*y*-axis) and the percentage of identified proteins (i.e., genes) involved in the respective pathway (*x*-axis) in each study group. Asterisk (*) denotes the highest group percentage in each analysed pathway.

**Figure 4 biomedicines-10-01522-f004:**
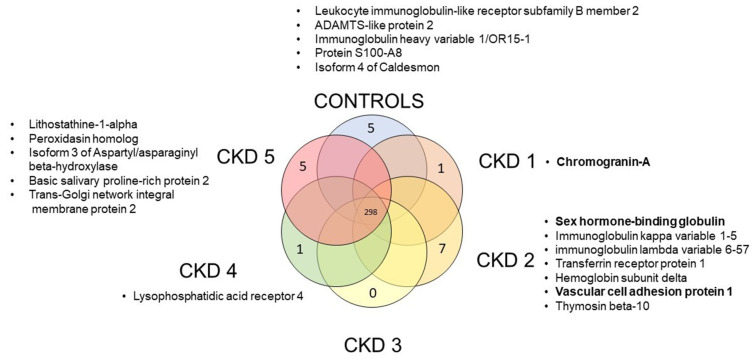
Venn diagram showing the number of proteins found in all study groups and proteins found only in one group. Specific proteins for each group are listed in boxes adjacent to the respective group. Proteins found to be potentially biologically relevant in CKD (according to the previously published literature) are shown in bold.

**Figure 5 biomedicines-10-01522-f005:**
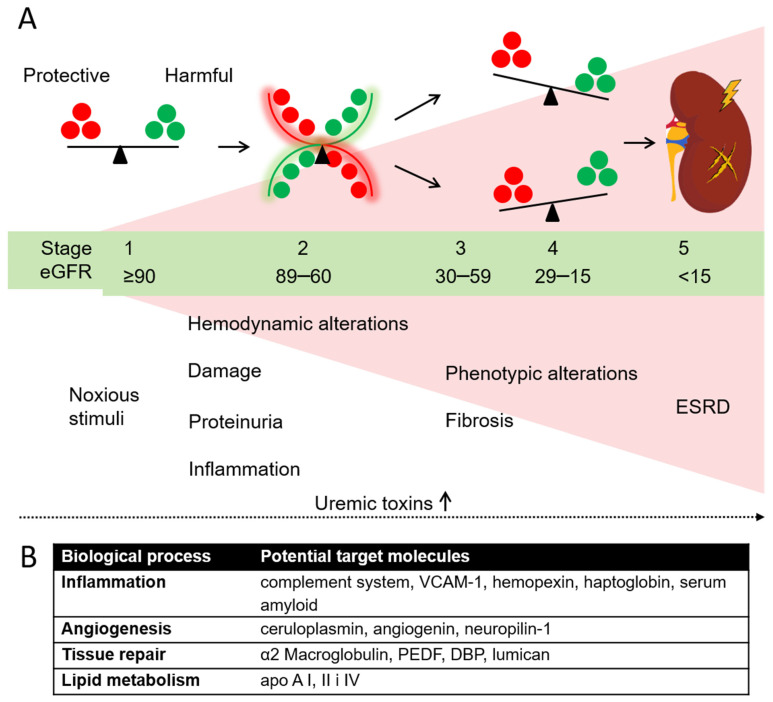
(**A**) Hypothetical presentation of Stage 2 CKD as a “tipping point” in further disease progression. (**B**) Proposed target proteins accompanied by a specific biological process significantly different in plasma of CKD patients stage 2 of disease as compared to the other CKD stages and healthy control group.

**Table 1 biomedicines-10-01522-t001:** Participants’ characteristics at the time of plasma sampling, categorized according to the stage of chronic kidney disease. Gender, comorbidities and underlying diseases are reported as number of participants (percentage). Age, body mass index and serum creatinine are reported as mean ± standard deviation. Proteinuria is reported as median (first and third quartile). Patients with hyperlipidemia were on statin therapy at the time of plasma sampling.

Group	Healthy	CKD Stage 1	CKD Stage 2	CKD Stage 3	CKD Stage 4	CKD Stage 5
N	15	15	15	15	15	15
Age (years)	42.9 ± 9.4	36.7 ± 15.2	54.5 ± 18.2	56.5 ± 14.5	60.5 ± 16.1	60.7 ± 15.4
Gender: N (%) female	7 (46%)	8 (53%)	7 (46%)	4 (26%)	9 (60%)	6 (40%)
BMI (kg/m^2^)	24.3 ± 2.7	25.1 ± 4.7	28.7 ± 6.43	25.7 ± 5.22	26.3 ± 4.06	28.3 ± 5.23
Serum creatinine (μmol/L)	72.7 ± 10.9	74.1 ± 14.3	88.7 ± 15.2	158 ± 23.7	237 ± 52.2	476 ± 136
eGFR (mL/min/1.73 m^2^)	104 ± 10.1	102 ± 10.2	74.7 ± 8.8	39.3 ± 7.2	21.7 ± 4.7	10.3 ± 2.3
Proteinuria (g/24 h)	0.2 (0–0.5)	0.36 (0.2–1.5)	0.7 (0.3–1.3)	0.49 (0.4–1.2)	0.43 (0.4–3.1)	1.88 (0.7–3.2)
BUN (mmol/L)	5 ± 1	5 ± 1.3	6.2 ± 1.6	10.5 ± 3.3	16.6 ± 4.9	24.3 ± 4.9
Serum cholesterol (mmol/L)	4.4 ± 0.6	5.4 ± 2.5	4.2 ± 0.8	4.5 ± 0.8	5 ± 1.4	5 ± 1.7
Serum triglycerides (mmol/L)	1.1 (0.7–1.4)	1 (0.8–1.2)	1.4 (1–1.8)	1.7 (1.2–2)	2 (1.6–3.4)	1.6 (1.4–2.8)
Serum HDL (mmol/L)	1.6 ± 0.3	1.6 ± 0.4	1.3 ± 0.4	1.3 ± 0.4	1.1 ± 06	1.1 ± 0.3
Serum LDL (mmol/L)	2.4 ± 0.5	3.3 ± 2.4	2.2 ± 0.7	2.1 ± 0.8	2.8 ± 1.4	2.7 ± 1.3
Serum albumin (g/L)	44 ± 3	39 ± 6	38 ± 3	40 ± 3	37 ± 6.6	40 ± 4
Comorbidities						
Hypertension	0	7 (47%)	14 (93%)	13.(86%)	15 (100%)	15 (100%)
Diabetes	0	0	3 (20%)	4 (26%)	5 (33%)	2 (13%)
Smoker	2 (13%)	2 (13%)	1 (7%)	6 (40%)	3 (20%)	8 (53%)
Hyperlipidemia	0	5 (33%)	12 (80%)	10 (67%)	12 (80%)	9 (60%)
Atherosclerosis	0	0	3 (20%)	4 (26%)	5 (33%)	8 (53%)
Underlying Disease						
Primary glomerular KD	0	13 (86%)	15 (100%)	9 (60%)	5 (33%)	6 (40%)
Hypertensive/atherosclerotic KD	0	0	0	2 (13%)	5 (33%)	2 (13%)
Autosomal dominant polycystic KD	0	1 (7%)	0	0	0	1 (7%)
Diabetic KD	0	0	0	0	1 (7%)	1 (7%)
Other specific cause	0	1 (7%)	0	2 (13%)	2 (13%)	2 (13%)
Unknown	0	0	0	1 (7%)	2 (13%)	3 (20%)

BMI—body mass index; BUN—blood urea nitrogen; CKD—chronic kidney disease; eGFR—estimated glomerular filtration rate; HDL—high density lipoprotein; KD—kidney disease; LDL—low density lipoprotein; N—number of participants.

**Table 2 biomedicines-10-01522-t002:** Relevant articles pertaining to notable group-specific proteins and its link with the degree of kidney disease.

Reference	Outlier Protein Name	N	Hospital Admission Disease	Study Groups	Tested Sample	Main Points and Conclusions
Po-Tseng Lee et al., 2021 [16]	VCAM-1	51	Peripheral arterial disease (PAD)	3 groups:A/normal kidney function; B/CKD; C/HD	serum/arterial tissue	- Link with underlying kidney disease - The HD group had a higher Concentration of VCAM-1 than the other two groups - Serum level and the tissue expression of VCAM-1 were significantly higher in PAD patients with advanced kidney disease
Gasparin et al., 2020 [17]	VCAM-1	62	Systemic lupus erythematosus (SLE)	2 groups:A/without active lupus nephritis (LN); B/with active LN	urine	VCAM-1 level was elevated in patients with active compared to inactive LN
Y Jia et al., 2021 [18]	VCAM-1	22	Diabetes mellitus	2 groups:A/Control; B/Diabetic kidney disease (DKD)	renal tubular cells, infiltrated immune cells	VCAM1 expression was upregulated in renal tubular cells, which might interact with infiltrated immune cells, thus promoting fibrosis.
Tramonti G et al., 2009 [19]	CGA	102	Kidney related disease	Patients with different values of GFR	serum	CGA accumulates in the blood in renal failure
Bech PR et al., 2012 [20]	CGA	147	Kidney related disease	2 groups: A/normal kidney function; B/CKD	plasma	CGA accumulates in the blood in renal failure
Yu et al., 2022 [21]	CGA	219	Type 2 diabetes mellitus (T2DM)	3 groups of patients with DN based on their urine albumin to creatinine ratios	serum	Serum CgA increased gradually with the degree of DN
Zhang H 2022 [22]	SHBP	5027	Screening for methabolic disease and other risk factors	Measurement of SHBP and testing correlation with eGFR	Serum, plasma	Lower serum SHBG levels were significantly associated with lower eGFR

N—number of participants; HD—hemodialysis; VCAM-1—Vascular Cell Adhesion Molecule; CGA—chromogranin A; SHBP—sex hormone-binding globulin; DN—diabetic nephropathy.

## Data Availability

The data presented in this study are openly available at ProteomeXchange Consortium via the PRIDE partner repository with the dataset identifier PXD033427.

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
