# Peer review of "Stage II of Chronic Kidney Disease—A Tipping Point in Disease Progression?"

_biomedicines, 2022, doi:10.3390/biomedicines10071522_

Round 1

Reviewer 1 Report

Overview

The manuscript reports results from a cross-sectional plasma proteomic study from patients with CKD stage 1-5. The study is observational in nature and concludes by suggesting a hypothesis regarding stage 2 CKD as an inflection point for progression. Although 90 patient samples were included in the study, samples were pooled so that individual variability cannot be determined. Pooling dampens initial enthusiasm but the study does provide data to suggest CKD stage changes in the plasma proteome if the authors can clarify the statistical methods. In fact, the methods section needs attention if the study is to be considered repeatable. All study limitations are included in a section of the discussion.

Minor comments

The author used the word “expression” like many previously published studies, but the word “expression” should be reserved to refer to gene expression. The study measured protein abundance, which is a result of gene expression, but the body does not express proteins.

Major Comments

Abstract. Please change this sentence “We performed a cross-sectional shotgun-proteomic study of plasma proteome across CKD stages…” to: We performed a cross-sectional shotgun-proteomic study of pooled plasma across CKD stages…”.

Introduction. The idea that CKD stage 1 or 2 is critical to progression has been discussed in the context of albuminuria (see  Ron T. Gansevoort, Paul E. de Jong. The Case for Using Albuminuria in Staging Chronic Kidney Disease, JASN Mar 2009, 20 (3) 465-468; DOI: 10.1681/ASN.2008111212). This study or similar studies should be included in the introduction because of its relevance to “key events” that promote CKD progression to ESRD.

Methods, Study Participants and study outline: The authors state:  “Participants with CKD had underlying glomerular disease.”. How was glomerular disease determined, and more importantly was glomerular disease determined even for patients with CKD stage 1?

Methods, Data Analysis. The Uniprot Proteome version should be included in the methods. E.g. Proteome ID: UP000005640, Release 2022_02

Methods, Data Analysis. Search parameters are missing. Please indicate whether full trypsin digest was utilized for in silico search. Also include modifications included in the search. This information may have been provided in PXD033427 submission, but this identifier is not public and could not be access in Proteomexchange.

Methods, Data Analysis. It is unclear how proteins were quantified. MS1, spectral count, emPAI, NSAF?

Methods, Data Analysis: Trend analysis needs to be described as does a definition for classifying proteins as significant. Its unclear how the authors can state a protein is significantly changing over stage because significance needs to be defined as P<0.0?.

Author Response

Point by point response:

Reviewer 1:

We wish to thank the Reviewer for helping us improve the quality of our manuscript and helping us convey our research findings in a more clear and transparent manner. Prompted by Reviewer 1’s remarks we have extensively revised the Materials and methods and Discussion sections.

The manuscript reports results from a cross-sectional plasma proteomic study from patients with CKD stage 1-5. The study is observational in nature and concludes by suggesting a hypothesis regarding stage 2 CKD as an inflection point for progression. Although 90 patient samples were included in the study, samples were pooled so that individual variability cannot be determined. Pooling dampens initial enthusiasm but the study does provide data to suggest CKD stage changes in the plasma proteome if the authors can clarify the statistical methods. In fact, the methods section needs attention if the study is to be considered repeatable. All study limitations are included in a section of the discussion.

We thank the Reviewer for their remarks which helped us improve the quality of our paper. We have rewritten and expanded the Materials and methods section on data analysis in order to present the research design more clearly. We have further emphasized that sample pooling limits interindividual variability detection, but it rather offers a “biological average” and that this research is focused on initial proteome screening.

Minor comments

The author used the word “expression” like many previously published studies, but the word “expression” should be reserved to refer to gene expression. The study measured protein abundance, which is a result of gene expression, but the body does not express proteins.

We thank the Reviewer, we corrected this.

Major Comments

Abstract. Please change this sentence “We performed a cross-sectional shotgun-proteomic study of plasma proteome across CKD stages…” to: We performed a cross-sectional shotgun-proteomic study of pooled plasma across CKD stages…”.

We thank the Reviewer, this has now been corrected.

Introduction. The idea that CKD stage 1 or 2 is critical to progression has been discussed in the context of albuminuria (see  Ron T. Gansevoort, Paul E. de Jong. The Case for Using Albuminuria in Staging Chronic Kidney Disease, JASN Mar 2009, 20 (3) 465-468; DOI: 10.1681/ASN.2008111212). This study or similar studies should be included in the introduction because of its relevance to “key events” that promote CKD progression to ESRD.

We thank the Reviewer, we have added the following text and reference in the Introduction section which we revised: “Gansevoort and de Jong argue that determination of proteinuria in stage 1 and 2 of CKD might be a relevant indicator of future progression to end-stage renal disease (ESRD). Furthermore, as they argue that albuminuria might be an early marker of renal impairment - patients with detected albuminuria without eGFR impairment might benefit from cardioprotective and renoprotective therapies.”

Methods, Study Participants and study outline: The authors state:  “Participants with CKD had underlying glomerular disease.”. How was glomerular disease determined, and more importantly was glomerular disease determined even for patients with CKD stage 1?

We thank the Reviewer for this inquiry, which prompted us to improve the study participants and study outline section to which we have now added the following: “The majority of participants with CKD suffered from underlying primary glomerular disease objectified by biopsy, including patients with stage 1 of CKD. Detailed distribution of underlying kidney diseases for each CKD stage is presented in Table 1.”

Methods, Data Analysis. The Uniprot Proteome version should be included in the methods. E.g. Proteome ID: UP000005640, Release 2022_02

We thank the Reviewer for pointing this out to us, this has now been corrected.

Methods, Data Analysis. Search parameters are missing. Please indicate whether full trypsin digest was utilized for in silico search. Also include modifications included in the search. This information may have been provided in PXD033427 submission, but this identifier is not public and could not be access in Proteomexchange.

We thank the reviewer for this remark, we have fully revised the “Data analysis” section of our manuscript in order to further clarify this point.

Methods, Data Analysis. It is unclear how proteins were quantified. MS1, spectral count, emPAI, NSAF?

We thank the Reviewer for their remark. This has now been corrected by adding the following: “The search parameters were: full trypsin digest, missed cleavages max. 2, dynamic modifications: Met oxidation, Asn and Gln deamination, dynamic protein N-terminal modifications: Met-loss, acetylation, Met-loss and acetylation. Label-free quantification was performed to determine the relative protein abundances, which were calculated from MS1 peptide intensities and normalized between samples”.

Methods, Data Analysis: Trend analysis needs to be described as does a definition for classifying proteins as significant. Its unclear how the authors can state a protein is significantly changing over stage because significance needs to be defined as P<0.0?.

We thank the Reviewer for their question and for prompting us to further clarify this in our manuscript. We have now fully revised the Materials and methods section which states the research design and “data processing” more clearly.

We made an omission and referred to protein abundance (previously referred to as expression, which we have now corrected) difference as “significant” in several places. We realize that the word “significant” denotes derivation from formal statistical tests. We have now corrected this and do not refer to our individual observations as “significant”. We meant to say that the observable difference in protein abundance is possibly biologically relevant. We now made extensive effort to edit the entire manuscript and to clearly convey our findings, along with potential bias and study limitations.

We conducted a shot-gun study of pooled plasma samples in order to screen for potential biomarkers. Initial step in our data processing was to determine weather each protein was detected in each group in a binary (yes/no) fashion. Then, using protein abundance we plotted our data in heatmaps and in a semiquantitative fashion analyzed our results and performed literature search for the identified proteins, Our literature search among the identified proteins then selected potential biologically relevant proteins. This process was done by manual curation and is prone to human-error and publication bias. We have now further stressed this in our Disucssion section.

To briefly summarize, we acted on this remark in three ways in order to improve our manuscript: a) we revised the Materials and methods section; b) we did not to refer to our observations as “significant” regarding to the findings where no statistical tests were performed; c) we further stressed the limitations of our study in the Discussion section.

Reviewer 2 Report

The authors properly identified the need and opportunity emerging from the existing literature for this kind of investigation. They professionally designed and conducted the measurements, which they clearly and comprehensively described in the Methods section of the manuscript. I must admit the presented style of writing confirms the authors’ experience and competence. Then, from the initial analysis, they received a large amount of data and they had to cope with the challenge and in general they had chosen proper techniques and tools. The authors have also done a very wide and comprehensive overview of the relevant literature. The whole work may be considered an interesting and inspiring example of implementation of the proteomic study in the selected branch of medicine. However, there are some observations that might somewhat mitigate the overall enthusiastic opinion. The subjects were 90 patients divided into 6 groups, based mainly on their eGFR. The size of each group (N=15) seems – as the authors openly admit in the Discussion section – rather small to allow drawing general conclusions about given sub-population (of patients with given CKD-stage). The effect of possible large impact of some individual patients, who accidentally fell into given group, could have been magnified by the fact that there were different patients in each group. To avoid such situation the observation over some relevant time the patients with gradually deteriorating CKD would be much safer. The effect of some local “anomaly”, which may have emerged accidentally, could be indicated even in Table 1, like: a) for BMI and the CKD-2 group, b) proteinuria and CKD-2, c) cholesterol and CKD-1. The second problem might be – as the authors also admit – the pooling process, during which they irreparably lost information about differences between individual patients in given group. The presented analysis could be of really considerable value if we could predict (from the relevant biomarker) further evolution of the disease in a person. Naturally, it is not possible when we consider all patients at given CKD-stage as a monolithic group and do not trace their further history.

My impression is that the whole study – initially based on proper grounds and performed with a lot of effort - gave no spectacular output and, therefore, the authors were forced to provide the presented comparative discussion and to emphasize some selected values. So, I agree with them that their results require verification, and should be rather treated as inspiration for some other study than considered as fully valid and instructive observations. If I have missed some important point,  the authors should modify the explanation of their achievements. In particular, the contents of Figure 5 should be exploited in the text with more details and emphasis, to properly point out the importance of the work. The last paragraph seems to be abruptly shortened, ­ the last sentence looks awkward and references [71-75] are not cited.

Author Response

Point by point response:

Reviewer 2:

We wish to thank the Reviewer for helping us improve the quality of our manuscript and helping us convey our research findings in a more clear and transparent manner. Prompted by Reviewer 2’s remarks we were able to identify mistakes and omission in our manuscript and timely correct them; we have also been able to improve the Discussion section.

The authors properly identified the need and opportunity emerging from the existing literature for this kind of investigation. They professionally designed and conducted the measurements, which they clearly and comprehensively described in the Methods section of the manuscript. I must admit the presented style of writing confirms the authors’ experience and competence. Then, from the initial analysis, they received a large amount of data and they had to cope with the challenge and in general they had chosen proper techniques and tools. The authors have also done a very wide and comprehensive overview of the relevant literature. The whole work may be considered an interesting and inspiring example of implementation of the proteomic study in the selected branch of medicine. However, there are some observations that might somewhat mitigate the overall enthusiastic opinion.

            We thank the Reviewer for their observations.

The subjects were 90 patients divided into 6 groups, based mainly on their eGFR. The size of each group (N=15) seems – as the authors openly admit in the Discussion section – rather small to allow drawing general conclusions about given sub-population (of patients with given CKD-stage). The effect of possible large impact of some individual patients, who accidentally fell into given group, could have been magnified by the fact that there were different patients in each group. To avoid such situation the observation over some relevant time the patients with gradually deteriorating CKD would be much safer. The effect of some local “anomaly”, which may have emerged accidentally, could be indicated even in Table 1, like: a) for BMI and the CKD-2 group, b) proteinuria and CKD-2, c) cholesterol and CKD-1.The second problem might be – as the authors also admit – the pooling process, during which they irreparably lost information about differences between individual patients in given group.

We thank the Reviewer for their remark: we have further stressed these observations regarding study design in the study limitations subsection of the Discussion in order to ensure maximum transparency regarding our research and study design.

The limitations section now reads: “A) One of the main limitations of our pilot proteomics study is the relatively small sample pool we draw our conclusions from: the presented ideas should be more broadly validated. B) This study is cross-sectional in design, a prospective longitudinal study in the future might offer further validation. C) Additionally, the staging of CKD relies on an attempt at quantification of kidney function through eGFR and it is blind to the underlying cause of kidney failure. Stratification of patients according to primary kidney disease in future studies should distinguish whether these concepts are universal or disease-specific. D) Another important limitation is sample pooling, which was a deliberate choice that enabled us to focus on the most prominent pathways disrupted in CKD, however it inherently diluted out individual patient specificities. Therefore, only the most prominent regulators of CKD progression might have been identified, while weaker, but possibly important modulators might have been disregarded. Sample pooling might have also clouded the distinction among underlying kidney disorders, however, the action of sample pooling allowed for a clearer data analysis. It is also theoretically possible that specific overabundant proteome pattern of a single patient acted as a potential confounder that “outweighed” group from the “biological average”. E) To add, the fact that we purified our samples using a heparin-enriched column “blinded” us to one part of plasma proteome, however multiple growth factors and cytokines that have a relatively high heparin binding affinity, were thus concentrated [62-68]. F) Finally, it is important to note that these results are primarily interpreted by manual curation done according to literature search results. Therefore, data interpretation is prone to human error and publication bias. Furthermore, the comprehensive nature of mass spectrometry-based proteomics, uncovered abundant differences in plasma proteome profiles of CKD patients as compared to healthy people; therefore, for the sake of clarity and readability, we have omitted to comment on pathways clearly known to be disrupted in CKD, such as the insulin like growth factor binding proteins. All raw data is, however, available at the ProteomeXchange Consortium via the PRIDE partner repository to achieve maximum transparency, as mentioned in the Data analysis section.

Furthermore, we revised the Materials and methods section in order to convey our study design more clearly.

The presented analysis could be of really considerable value if we could predict (from the relevant biomarker) further evolution of the disease in a person. Naturally, it is not possible when we consider all patients at given CKD-stage as a monolithic group and do not trace their further history.

We thank the Reviewer for their observations. As the Reviewer stated, our study is cross-sectional in design and thus has limitations that are mentioned. We employed a cross-sectional study design with sample pooling in order to initially screen plasma proteome. We are fully aware that further validation studies with longitudinal design are necessary – we have thus stressed this further in the Discussion: In the limitations subsection we wrote: “This study is cross-sectional in design, a prospective longitudinal study in the future might offer further validation”; and in the final subsection: “However, further prospective studies with longitudinal design, including more participants and with adequate patient stratification are needed to confirm our hypothesis. These studies should, therefore, be the basis of further biomarker screening and future validation, which might ultimately have an impact on therapeutic strategies and potentially improve patient outcomes.”

My impression is that the whole study – initially based on proper grounds and performed with a lot of effort - gave no spectacular output and, therefore, the authors were forced to provide the presented comparative discussion and to emphasize some selected values. So, I agree with them that their results require verification, and should be rather treated as inspiration for some other study than considered as fully valid and instructive observations. If I have missed some important point,  the authors should modify the explanation of their achievements.

We thank the Reviewer for their remark. We further stressed the need for future studies in order to verify our results in the limitations subsection, as well as in the final subsection: “We thus hypothesize that stage 2 might be the potential “site” for biomarker discovery and for development of therapeutic strategies for halting CKD. However, further prospective studies with longitudinal design, including more participants and with adequate patient stratification are needed to confirm our hypothesis. These studies should, therefore, be the basis of further biomarker screening and future validation, which might ultimately have an impact on therapeutic strategies and potentially improve patient outcomes.”

In particular, the contents of Figure 5 should be exploited in the text with more details and emphasis, to properly point out the importance of the work.

We thank the Reviewer, we have revised parts of the Discussion section in order to, as Reviwer stated, further exploit Figure 5 in the text.

The last paragraph seems to be abruptly shortened, ­ the last sentence looks awkward and references [71-75] are not cited.

We thank the Reviewer for their remark, we have corrected the problem with the references.  Namely, our citation software did not recognize the references cited in the table. This has now been corrected.

The final section now reads:

The key message of our study is the observation of synchronous activation of cellular pathways related to inflammation and angiogenesis as well as tissue homeostasis and repair. In that regard, stage 2 seems to be a potential “tipping point” in CKD progression, a possible molecular milestone in natural disease progression, beyond which, hallmarks of disease, such as hemodynamic changes, proteinuria and inflammatory processes become increasingly evident both clinically and by proteomic analysis (Figure 5). Different biological processes known to be important for CKD progression are reflected in several proteins identified as potential target molecules (Figure 5). We thus hypothesize that stage 2 might be the potential “site” for biomarker discovery and for development of therapeutic strategies for halting CKD. However, further prospective studies with longitudinal design, including more participants and with adequate patient stratification are needed to confirm our hypothesis. These studies should, therefore, be the basis of further biomarker screening and future validation, which might ultimately have an impact on therapeutic strategies and potentially improve patient outcomes.

Round 2

Reviewer 2 Report

I have no further objections. I hope this work will inspire the researchers to arrange for the relevant longitudinal studies, and finally adequate biomarkers will be found.